# Quinuclidinium salt ferroelectric thin-film with duodecuple-rotational polarization-directions

Yu-Meng You[1,2], Yuan-Yuan Tang[1,2], Peng-Fei Li[1,2], Han-Yue Zhang[1,2], Wan-Ying Zhang[1,2], Yi Zhang[1,2], Heng-Yun Ye[1,2], Takayoshi Nakamura[3] & Ren-Gen Xiong[1,2]

Ferroelectric thin-films are highly desirable for their applications on energy conversion, data storage and so on. Molecular ferroelectrics had been expected to be a better candidate compared to conventional ferroelectric ceramics, due to its simple and low-cost film-processability. However, most molecular ferroelectrics are mono-polar-axial, and the polar axes of the entire thin-film must be well oriented to a specific direction to realize the macroscopic ferroelectricity. To align the polar axes, an orientation-controlled single-crystalline thin-film growth method must be employed, which is complicated, high-cost and is extremely substrate-dependent. In this work, we discover a new molecular ferroelectric of quinuclidinium periodate, which possesses six-fold rotational polar axes. The multi-axes nature allows the thin-film of quinuclidinium periodate to be simply prepared on various substrates including flexible polymer, transparent glasses and amorphous metal plates, without considering the crystallinity and crystal orientation. With those benefits and excellent ferroelectric properties, quinuclidinium periodate shows great potential in applications like wearable devices, flexible materials, bio-machines and so on.

[1] Ordered Matter Science Research Center, Southeast University, Nanjing 211189, P. R. China. [2] Jiangsu Key Laboratory for Science and Applications of Molecular Ferroelectrics, Southeast University, Nanjing 211189, P. R. China. [3] Research Institute for Electronic Science, Hokkaido University, Sapporo 001-0020, Japan. Correspondence and requests for materials should be addressed to Y.-M.Y. (email: youyumeng@seu.edu.cn) or to R.-G.X. (email: xiongrg@seu.edu.cn).

Interest in ferroelectrics with switchable spontaneous polarizations under external electric fields has been driven by the extensive applications in ferroelectric random access memories, piezoelectric sensors, nonlinear optical devices, capacitors and so on[1–3]. With the increasing demand on thin-film devices and printed electronics, the complicated film-processing procedures of inorganic ferroelectrics hampered their development[4–6], and increasing attentions have been attracted to molecular ferroelectrics due to their simple full-solution processing and low-cost film-making. In the past decade, studies on molecular ferroelectrics achieved great improvement and a series of new compounds have been discovered with spontaneous polarization and transition-temperature comparable or superior to those of inorganics[7–11].

Despite those advantages, in the aspect of practical applications, realization of macroscopic ferroelectricity on molecular thin-films is still challenging. As most of molecular ferroelectrics are mono-polar-axial, which means the polarization can only be switched to two specific directions, the thin-film must be prepared with specific crystallographic orientation to maximize the effective polarization. With such a concern, one could only use either well-controlled epitaxial large area growth or polycrystalline film growth where polarization direction is randomized but the effective polarization intensity is sacrificed. For example, our group has reported a molecular ferroelectric thin film of imidazolium perchlorate, showing high spontaneous polarization and superior electromechanical coupling, but the switchable spontaneous polarization can only be realized in micrometer-scale on the polycrystalline thin-film due to its mono-polar-axial nature[12,13]. Encouragingly, a recent discovery of ferroelectricity in plastic crystal, quinuclidinium perrhenate, offers a ray of hope to solve this issue[14]. To be more specific, the majority of plastic crystals often belong to the high-symmetry cubic point groups in the high-temperature paraelectric phase and have more than three polarization axes. Although the rotation of polarization is exciting in quinuclidinium perrhenate, the narrow temperature-window (345–367 K) of ferroelectric phase strictly limits its application potential.

Herein we discover a compound of quinuclidinium periodate (**1**), which experiences a ferroelectric phase transition at 322 K. At room temperature, **1** possesses six-fold polar axes with a spontaneous polarization of 7.71 $\mu C\,cm^{-2}$. **1** crystallizes in the orthorhombic space group $Pmn2_1$ at room temperature and in the cubic space group $Pm\bar{3}m$ at high temperature. Therefore, similar to the orthorhombic ferroelectric phase of $BaTiO_3$, there are six polarization axes existing in the crystal of **1**. Comparing to the recently reported quinuclidinium perrhenate, **1** is a room-temperature molecular ferroelectric compound with more polar axes and higher spontaneous polarization (a detailed comparison can be found in the Supplementary Table 1). Taking advantages of the six-fold polar axes and 12 polarization states, we have successful demonstrated macroscopic ferroelectric polarization reversal on thin-film samples prepared by simple aqueous solution process. More importantly, such ferroelectricity does not require any orientation-controlled growth on specific substrate, which provide **1** great feasibility on making thin-film devices on various substrates, including flexible polymers. Such realization of macroscopic ferroelectricity on polycrystalline thin-films makes **1** a molecular ceramic-like ferroelectric. This discovery might open a new avenue for the device applications of molecular ferroelectrics, especially in flexible devices, wearable electronics, micromechanics and so on.

## Results

**Crystal structures.** As a characteristic of ferroelectrics, **1** has a high-temperature paraelectric phase and a low-temperature ferroelectric phase (for crystal data, see Supplementary Data 1 and 2, and Supplementary Table 2). At room-temperature, **1** is in the low-temperature phase (LTP), which is in the orthorhombic space group $Pmn2_1$, $a = 9.099(9)$, $b = 6.059(6)$, $c = 9.240(9)$ Å. For temperature above the phase-transition temperature ($T_c$), **1** is in the high-temperature phase (HTP), which is in the cubic space group $Pm\bar{3}m$. $a = 6.437(8)$ Å. The relationship of the cell constants is $\mathbf{a}_{LTP} = \mathbf{a}_{HTP} - \mathbf{c}_{HTP}$, $\mathbf{b}_{LTP} = \mathbf{b}_{HTP}$, $\mathbf{c}_{LTP} = \mathbf{a}_{HTP} + \mathbf{c}_{HTP}$. To compare structures of HTP and LTP, perspective views of the two phases are shown in Fig. 1 with same crystal orientation.

In the HTP, both the cation and the anion are located on the special sites, $m\bar{3}m$, as shown in Fig. 1b. The intermolecular symmetries are not consistent with those of the crystallographic symmetry. The crystallographic symmetry can only be satisfied by disorder of the cation and anion. The modelled molecular structure can be understood as the average of multiple orientations. Therefore, the multiple disorder in the HTP might be dynamical, which is characterized by heavy orientational or displacive disorder of molecules[15]. In our case, due to the

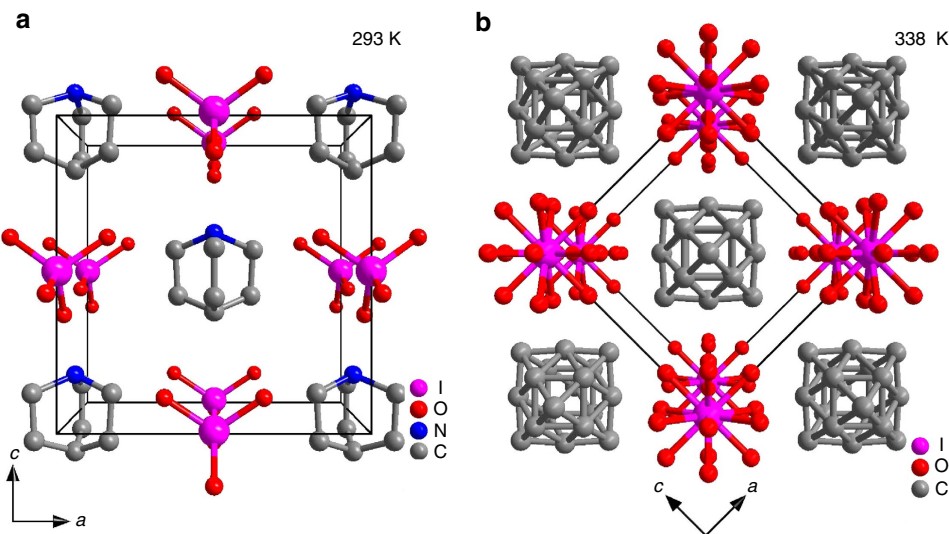

**Figure 1 | Comparison of crystal structures of 1 in the HTP and LTP.** (**a**) Perspective view of the crystal structure in the LTP. (**b**) Perspective view of the crystal structure in the HTP. The same orientation of the structures was chosen for comparison purposes. H atoms were omitted for clarity.

spherical structure of the cation and anion, only severely orientational disorder was observed in the HTP of **1**, as illustrated in Fig. 1b. For temperature below $T_c$, such disorder is frozen due to the decreasing of thermal perturbation, as shown in Fig. 1a. The orientations of the cations are the same, and there is obvious displacement between the positive charge (located on the N atom) and the negative charge (located on the centre of the $IO_4^-$ anion), which leads to a ferroelectric spontaneous polarization. On the basis of a point charge model, the spontaneous polarization is estimated to be $7.71\,\mu C\,cm^{-2}$ (Supplementary Note 1 and Supplementary Table 3). To exhibit ferroelectric bistability, the cation should reorient upon applying a backward electric field. The reorientation involves relatively great amplitude of the motion of the cation and anions in comparison of the displacements in inorganic ferroelectric materials.

**Ferroelectric phase transition.** To examine the properties of the paraelectric–ferroelectric structural phase transition in **1**, we performed differential scanning calorimetry (DSC), dielectric and second harmonic generation (SHG) measurements. The DSC curves clearly indicate a phase transition at about $T_c = 322\,K$ (Supplementary Fig. 1). The sharp peak and the large temperature hysteresis (7 K) of the thermal anomalies suggest that phase transition is of first-order type[16]. The temperature dependence of the real part of the dielectric permittivity ($\varepsilon'$) of **1** was measured on powdered sample with various frequencies. As illustrated in Fig. 2a, the temperature-dependent dielectric permittivity displays a distinct dielectric anomalies with the peak value of 186 at 1 MHz upon cooling, which is the characteristic feature of paraelectric–ferroelectric phase transition. The breakdown of centrosymmetry during the phase transition is verified by monitoring the temperature-dependent SHG response. From

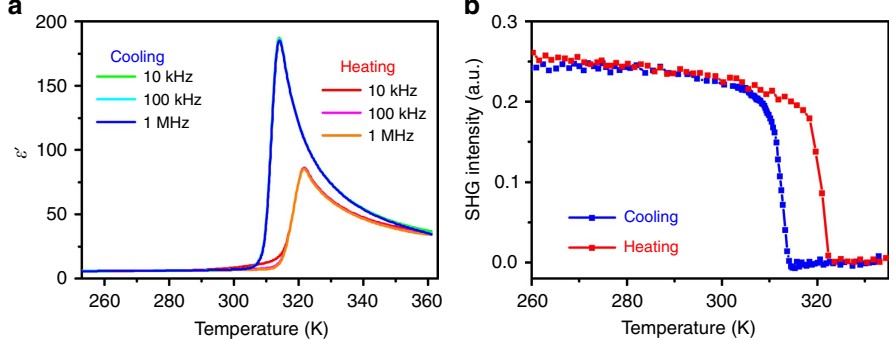

**Figure 2 | Dielectric and SHG responses of 1.** (**a**) Real part of the dielectric permittivity as a function of temperature at different frequencies. (**b**) Temperature-dependent SHG response, revealing a symmetry-breaking phase transition at around $T_c = 322\,K$.

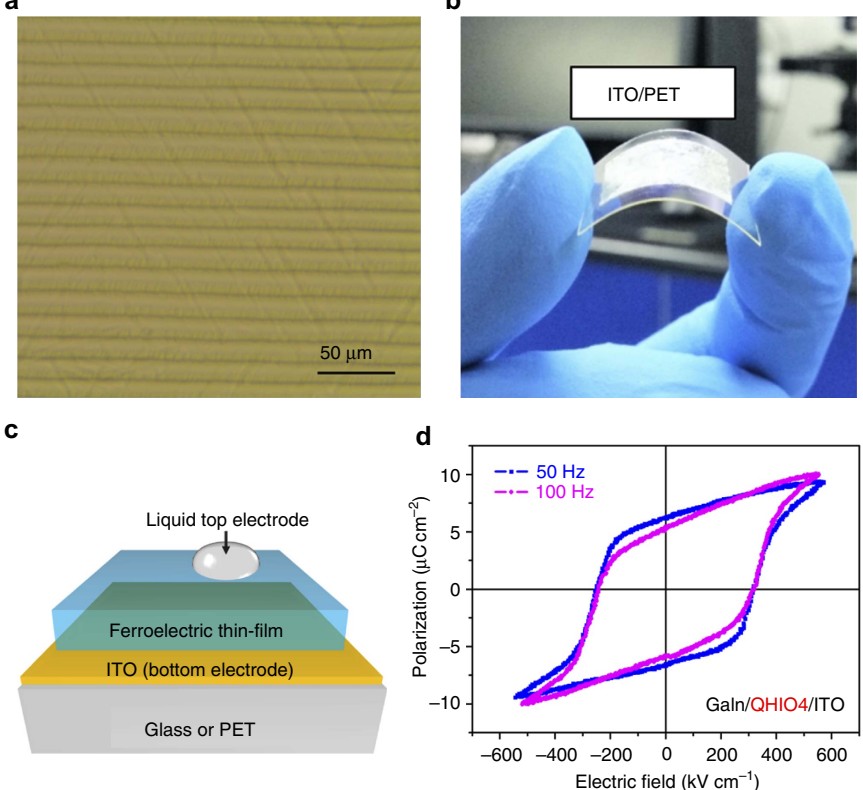

**Figure 3 | Properties of the thin-film of 1.** (**a**) Optical microscope image of the thin-film of **1** deposited on ITO-coated glass. (**b**) Photo of thin-film of **1** on a flexible substrate. (**c**) Illustration of thin-film device for ferroelectric characterization. (**d**) Ferroelectric hysteresis loops obtained on thin-film of **1** on ITO/glass substrate.

Fig. 2b, a strong SHG signal was perceptible at low temperature. As increasing temperature, the SHG intensity decreases gradually and finally vanished at around $T_c$, suggesting that **1** is centrosymmetric in HTP and noncentrosymmetric in LTP.

**Ferroelectric thin-films.** The ferroelectric thin-film was prepared by a simple solution-based method. As shown in Fig. 3a, the as-prepared thin-films has large area, high uniformity and high-coverage. Benefited from the full-solution processing,

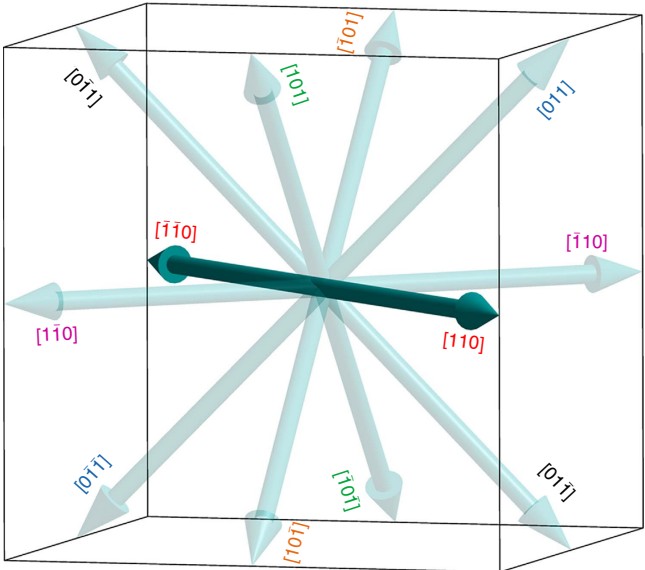

**Figure 4 | Equivalent polarization directions in 1.** The 12 equivalent polarization directions in the ferroelectric phase were calculated according to the symmetry of its cubic paraelectric phase. 60°, 90°, 120° and 180° domain angle can be theoretically predicted.

thin-films of **1** can be prepared on a wide range of substrates, including ITO glass, glass, $SiO_2/Si$, single-crystalline silicon, steel plate and so on as shown in Supplementary Figs 2 and 3. From the optical microscopic image shown in Fig. 3a, periodical dendritic pattern can be observed on the thin-film surface which are in the scale of micrometres. The surface shows very good coverage with no visible pin-hole in the entire view area $(0.1 \times 0.1\,\text{mm})$, which was also confirmed by atomic force microscopy, as shown in Supplementary Fig. 4. The low-processing temperature of **1** also allows making flexible thin-film by employing substrate of ITO-coated polyethylene terephthalate (PET), as shown in Fig. 3b.

**Macroscopic ferroelectricity on polycrystalline thin-films.** ITO-coated glass was chosen as substrate to characterize the ferroelectric properties of the thin-film with the conductive ITO coating serving as bottom electrode. Drops of liquid GaIn eutectic (diameter of $\sim 0.5\,\text{mm}$) was applied as the top electrode to form a capacitor architecture (GaIn/ferroelectric thin-film/ITO), as illustrated in Fig. 3c. A standard Sawyer-Tower circuit was employed to measure the polarization as a function of external electric field. As shown in Fig. 3d, rectangular *P-E* hysteresis loops at different frequencies are well presented, affording indispensable proof for the room temperature ferroelectricity of **1**. From *P-E* hysteresis loops, the coercive field ($E_c$) is extracted as $255\,\text{kV cm}^{-1}$, which is significantly larger than that of common molecular ferroelectrics (typical $5 \sim 20\,\text{kV cm}^{-1}$). In order to obtain a proper hysteresis loop, the film thickness has to be very small ($< 5\,\mu\text{m}$) so the applied bias voltage can overcome the large coercive potential. Besides, the film of **1** shows a considerable remnant polarization with a value of $6.7\,\mu\text{C cm}^{-2}$ at 40 Hz, which is slightly smaller than that theoretically predicted $7.71\,\mu\text{C cm}^{-2}$ with point charge model and $9.01\,\mu\text{C cm}^{-2}$ with Berry phase calculation (for details, see Supplementary Fig. 5, Supplementary Method and Supplementary Refs 1–5). In addition, similar ferroelectricity can be obtained from a film

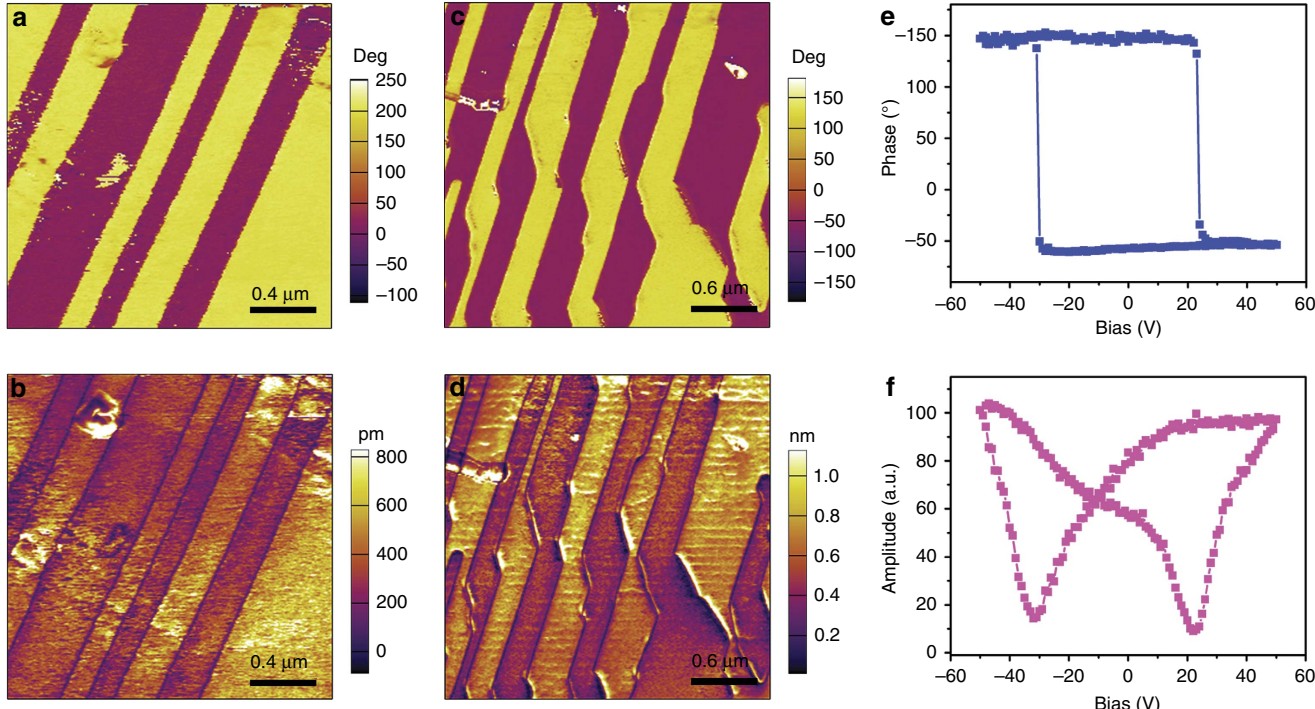

**Figure 5 | PFM images for the thin-film of 1.** (**a,c**) Vertical PFM phase images. (**b,d**) Vertical PFM amplitude images. (**e,f**) Phase and amplitude signals as functions of the tip voltage for a selected point, showing local PFM hysteresis loops.

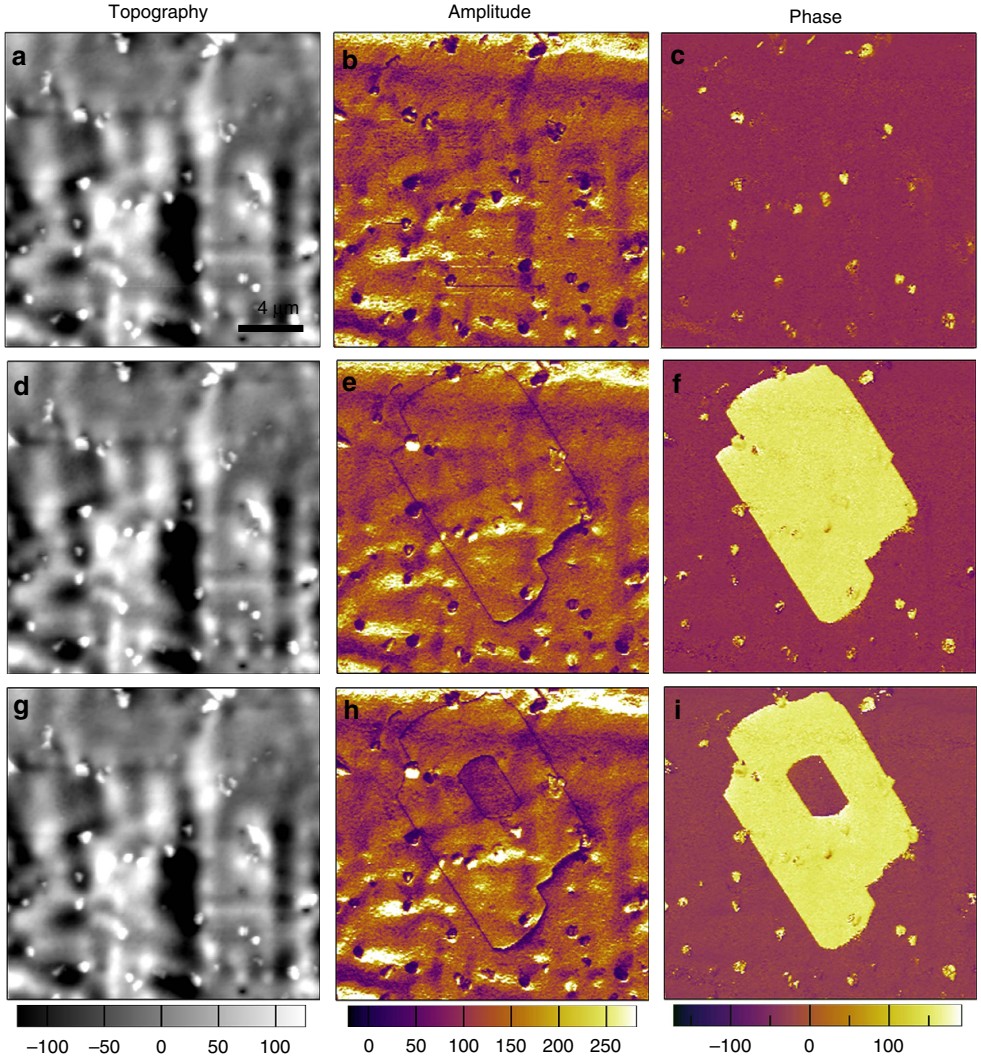

**Figure 6 | Polarization reversal investigated by PFM.** The panels in each row are arranged as the sequence: the topographic image (left), the vertical PFM amplitude image (middle) and the phase image (right) of the surface. (**a**–**c**) PFM images of the as-prepared thin-film surface. (**d**–**f**) PFM images after the first polarization writing (switching) with positive tip-bias of $+20$ V. (**g**–**i**) PFM images after the second polarization writing in a smaller area with tip-bias of $-25$ V. The yellow and red contrast in phase images indicate the regions with polarization oriented upward and downward, respectively. The temperature-dependent PFM imaging technique was also employed to visualize the evolution of domain structures during the paraelectric–ferroelectric phase transition, as shown in Supplementary Fig. 9. All the panels have the same scale bar.

deposited on a flexible ITO-PET substrate as shown in Supplementary Fig. 6. The experimentally obtained polarization value appears lower than the theoretical estimation, due to the polycrystalline nature of the thin-film sample. Normally, the theoretical calculation is performed based on a single-crystal model. However, in a thin-film sample, the polarization direction of each grain may not be strictly along the normal direction of the film, resulting a slightly smaller effective spontaneous polarization comparing to that of a single-crystal sample.

**Twelve-fold polarization direction.** Normally, to obtain a good *P-E* hysteresis loop on ferroelectrics, especially on molecular ferroelectrics which are mostly mono-polar-axial, a good quality single-crystalline sample is necessary, and the two electrodes should be deposited along the polar axes. While, in our case, on a device based on polycrystalline thin-film of **1** with an arbitrary top-bottom-electrodes configuration, it is surprising that one can still observe a good ferroelectric behaviour. The reason of such independency on

crystallographic orientation and substrate is the unique multi-polar characteristic of **1**.

According to the change of symmetry during phase-transition, the crystal of **1** belongs the species of $m\bar{3}mFmm2$ (Aizu notation) among the 88 ferroelectric species[17]. The ferroelectric LTP adopts the space group $Pmn2_1$ (point group: $mm2$), whose polarization direction is confined to [001]-direction by the crystallographic symmetry (Supplementary Fig. 7). Considering its cubic paraelectric phase with space group $Pm\bar{3}m$ (point group: $m\bar{3}m$), the paraelectric-to-ferroelectric phase transition will give rise to the spontaneous polarization in [110]-direction of the cubic prototype phase, which is coincided with the [001]-direction of its room temperature orthorhombic ferroelectric phase (Supplementary Fig. 7). More importantly, there are 12 equivalent [110] directions in the cubic paraelectric phase, which means 12 possible polarization directions exist in ferroelectric LTP, as shown in Fig. 4.

**Ferroelectric domain structures.** Piezoresponse force microscopy (PFM) imaging has emerged as an effective characterization

tool to directly observe the statics and dynamics of ferroelectric domains at the nanometre scale[18–24]. Each PFM image can be characterized by the phase and amplitude parameters to provide information about the orientation of the domain polarization and the value of the piezoelectric coefficient, respectively. Figure 5 presents images constructed by vertical PFM amplitude and phase, which distinctly reveal two kinds of typical domain structures for the polycrystalline thin-film of **1**. The first type is stripe domain, like that in $BiFeO_3$ (BFO), as depicted in Fig. 5a,b. The phase contrast indicates the oppositely oriented domains with upward and downward polarizations, and the dark hatched regions in the amplitude image obviously demonstrate the existence of domain walls, which correspond well to each other. The second type is shown in Fig. 5c,d, where a distorted the stripe domain was observed and the angle between two domain walls was always around 150°. Such a structure signifies the possible existence of non-180° domains. The phase image exhibits two oppositely oriented domains in the vertical component with a clear contrast, separated by the domain wall shown in the amplitude image. A detailed characterization on the non-180° domains, which employed PFM to image the vertical (out-of-plane) and lateral (in-plane) components of the piezoresponse can be found in Supplementary Fig. 8.

We then carried out PFM spectroscopy to study the local polarization switching behaviour and the local piezoresponse as a function of the bias tip voltage. From Fig. 5e,f, the characteristic hysteresis and butterfly loops can be observed on the film surface, which is a typical demonstration of switching the ferroelectric polarization. In addition, to direct visualize the domain switching process, we use PFM with DC tip-bias voltage to manipulate the local domain structure, as shown in Fig. 6. Firstly, the vertical PFM images were recorded on certain area of the as-prepared film surface (Fig. 6a–c). Subsequently, the centre part of that area was re-scanned with a DC tip-bias of $+20$ V, which is beyond the coercive potential for the film. In the phase image, the centre area with reversed polarization direction can be seen after the first polarization writing/switching, as shown in Fig. 6d–f. Also, the domain walls were clearly shown in the amplitude image match well with that in the phase image. Then, a smaller area was scanned again with negative DC tip-bias of $-25$ V, and series of PFM images were recorded and shown in Fig. 6g–i. From the phase image of Fig. 6i, the inner part of the yellow (upward polarized) region was switched to red (downward polarized), and similar domain wall appears in the amplitude image. Such polarization writing has directly demonstrated the switching capability of the thin-film disregarding its polycrystalline nature.

## Discussion

As summary, we report a new molecular ferroelectric compound, quinuclidinium periodate (**1**), with phase-transition temperature $T_c = 322$ K. It crystallizes in the orthorhombic space group $Pmn2_1$ at room temperature ferroelectric phase and the cubic space group $Pm\bar{3}m$ at high temperature paraelectric phase. Resultantly, accompanying the phase transition, 6 polar axes exist in ferroelectric LTP, leading to 12 different possible polarization directions. By analysing the crystal symmetries in two phases of **1**, angle of 60°, 90°, 120° and 180° between different polarization directions in the LTP can be deduced. With the help of multi-axial characteristic and high-quality thin-film processability on various substrates, including transparent and flexible ITO/PET substrate, macroscopic ferroelectricity is successfully demonstrated at room temperature. The measured spontaneous polarization is as high as $6.7\,\mu C\,cm^{-2}$, comparable to that of classical molecular ferroelectrics, such as triglycine sulfate (TGS, $3.8\,\mu C\,cm^{-2}$). Combining the above mentioned advantages,

**1** shows great potential in thin-film devices, due to its minimum requirements on crystallinity and specific substrate. The solution-based thin-film processing and the excellent ferroelectricity make it an ideal candidate for applications in next-generation flexible electronics.

## Methods

**Materials.** All reagents and solvents in the syntheses were of reagent grade and used without further purification. Quinuclidinium periodate (**1**) was prepared by slow evaporation of the ethanol solution under quinuclidine and periodate in a 1:1 molar ratio. Elemental analysis (%) calculated for $C_7H_{14}NO_4I$: C, 27.74; H, 4.66; N,4.62. Found (%) C, 27.72; H, 4.68; N, 4.66. The purity of the bulk phase was verified by the infrared spectrum (Supplementary Fig. 10) and powder X-ray diffraction (Supplementary Fig. 11). Single-phase grains of **1** were dissolved in purified water to form a solution with a solubility of about $200\,mg\,ml^{-1}$. With this solution, the thin film was deposited on ITO-coated glass and ITO-coated PET substrates, respectively.

**Thin-films on various substrates.** We take ITO (Indium Tin Oxide) -coated glass substrate as an example. The precursor solution of **1** was carefully spread on a freshly cleaned ITO glass (conductive ITO as the bottom electrode). With controlled substrate temperature and edge-pinned-crystallization, a uniform film consisting of continuous dendritic crystal with high coverage was obtained.

**Measurements.** DSC, SHG, dielectric and polarization-electric field (P–E) hysteresis loops measurements were described elsewhere[25]. Macroscopic ferroelectric properties were measured by a commercial Sawyer-Tower circuit (Premier II Ferroeclectric Tester).

Nanoscale polarization imaging and local PFM spectroscopy were carried out using a resonant-enhanced PFM (MFP-3D, Asylum Research). Conductive Pt/Ir-coated silicon probes (EFM-50, Nanoworld) were used for domain imaging and polarization switching studies. In PFM imaging, 1 V AC driving voltage was applied to measure the normal and shear responses, with the AC frequency set at the second resonant peak of cantilever-sample system (350 kHz for normal response and 710 kHz for shear response) to enhance the sensitivity.

**Data availability.** The structures have been deposited at the Cambridge Crystallographic Data Centre (deposition numbers: CCDC 1499452-1499453), and can be obtained free of charge from the CCDC via www.ccdc.cam.ac.uk/getstructures.

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

## Acknowledgements

This work was supported by 973 project (2014CB932103), the National Natural Science Foundation of China (21290172, 91422301, 21427801 and 91622113) and Jiangsu Natural Science Foundation (BK20160029).

## Author contributions

H.-Y.Z. and W.-Y.Z. prepared the samples. Y.-Y.T., P.-F.L. and Y.Z. characterized the properties. H.-Y.Y. determined the crystal structures. T.N., Y.-M.Y. and R.-G.X. wrote the manuscript. Y.-M.Y. and R.-G.X. designed and directed the studies.
