## [Peer Review File · Nature Communications]

Reviewers' comments:

Reviewer #1 (Remarks to the Author):

You et al. reported the ferroelectric properties of quinuclidinium periodate crystals and their thin films. Quinuclidinium periodate exhibits ferroelectric properties above room temperature. A thin film of quinuclidinium periodate, simply prepared without considering molecular orientation on the substrates, exhibits ferroelectric hysteresis because it possesses a six-fold rotational polar-axis. The structure and properties of the compound are well characterized using single crystal analysis, dielectric measurement, SHG, P–E curve, and piezoresponse force microscopy. A thin film of the quinuclidinium periodate with ferroelectric properties would be very useful in applications of quinuclidinium periodate.

I have several concerns listed as follows, which should be addressed.

The present study is closely related to the work by Harada et al. (ref.17), who reported directionally tunable ferroelectric crystals in quinuclidinium perrhenate. Therefore, a detailed comparison between the author's and Harada's work is necessary. Note that it would be helpful to readers if the authors clearly state the novelty of their work, especially, when compared with Harada's results.

In ref.17, Harada et al. raised a problem for normal molecular ferroelectric crystals: "restricted orientational control over molecular crystals has severely limited their applications in electronic devices." Therefore, they synthesized and characterized quinuclidinium perrhenate, in which the polarization axis can be easily changed to the desired direction through rotation of the constituent ionic molecules of the crystal. However, they have not investigated thin films. Therefore, the present study should focus more on the characterization of ferroelectric thin films.

The six-fold rotational polar-axis is another major concern. Single crystal analysis shows that the quinuclidinium periodate has a six-fold rotational polar-axis. This suggests that an identical P–E curve should be observed when an electric field is applied from any of the six directions; however, it is not directly demonstrated by experiment. The paper could be strengthened by performing P–E curve measurements using single crystal in at least two different directions.

Minor issues

Color and atoms should be correlated in Fig. 1.

Figures 2a and 2b should be plotted for the same temperature range, for example, between 280 and 360 K to enable comparison.

Color in Fig. 6 (Phase) appears to be red, and not purple.

In the Berry phase calculation section, " λ in Figure 3" should be revised to " λ in Figure S1."

It would be helpful to provide the definition of λ in Fig. S1.

Label for the y-axis should be added to Fig. S2.

"Heigt" in Fig. S7b should be revised to "Height."

Reviewer #2 (Remarks to the Author):

This work discovered the new molecular ferroelectrics of quinuclidinium periodate, which can be readily prepared on various substrates without the concern on the film crystallinity and crystal-orientation. The multi-axes (six polar axes) molecular ferroelectrics exhibits the dendritic growth pattern and 12-fold polarization direction. As a result, the thin films with switchable polarization possess great application potentials. There are a great many of outstanding points of this work, such as the simple preparation of aqueous solution processing and the flexible structure on ITO-PET electrode substrate. In addition, the micrometer scale thin-film can be fabricated on various substrates, including the flexible polymers with high uniformity. I strongly recommend this work for the publication.

There are two minor comments for the authors' consideration: 1) It may be helpful to add the rational of smaller remnant polarization (currently discussed in the supporting information as compared to the theoretically predicted one) into the main text. 2) The author may consider add a few sentences to discuss the strategies to reduce the coercive field of the films, though six polar axes of quinuclidinium periodate.

REVIEWERS' COMMENTS:

Reviewer #1 (Remarks to the Author):

The manuscript has been revised appropriately on the basis of two reviewers' comments. The manuscript is worthy for publication after addressing the following issues.

Minor issues

Color and atoms should be correlated in Fig. 1; N (blue) is not included in the revised Figure (Fig. 1).

It would be useful to include the elemental analysis (CHN) of 1 in the Materials section.

Reviewer #2 (Remarks to the Author):

The authors have addressed and revised the manuscript, which is ok for the publication.

Reviewers' comments:

Reviewer #1 (Remarks to the Author):

You et al. reported the ferroelectric properties of quinuclidinium periodate crystals and their thin films. Quinuclidinium periodate exhibits ferroelectric properties above room temperature. A thin film of quinuclidinium periodate, simply prepared without considering molecular orientation on the substrates, exhibits ferroelectric hysteresis because it possesses a six-fold rotational polar-axis. The structure and properties of the compound are well characterized using single crystal analysis, dielectric measurement, SHG, P-E curve, and piezoresponse force microscopy. A thin film of the quinuclidinium periodate with ferroelectric properties would be very useful in applications of quinuclidinium periodate.

Author's response:

We would like to thank the reviewer for his/her positive remark on the importance of our manuscript, and also for their suggestions to help us improve the quality of this manuscript as well as our future research indeed. Our point-to-point response are listed below in blue color.

I have several concerns listed as follows, which should be addressed.

The present study is closely related to the work by Harada et al. (ref.17), who reported directionally tunable ferroelectric crystals in quinuclidinium perrhenate. Therefore, a detailed comparison between the author's and Harada's work is necessary. Note that it would be helpful to readers if the authors clearly state the novelty of their work, especially, when compared with Harada's results.

Author's response:

We have added the comparison between those two compounds in the main text (page 4) featuring following points

- 1) The ferroelectric phase of **1** has a broader temperature range (< 322 K) than that of quinuclidinium perrhenate (345–367 K).
- 2) The remnant polarization of **1** (6.2–6.8 $\mu\text{C}/\text{cm}^2$) is larger than that of quinuclidinium perrhenate (3.5 $\mu\text{C}/\text{cm}^2$).
- 3) **1** has a significantly larger coercive field (255 kV/cm) than that of quinuclidinium perrhenate (2-5 kV/cm).

A detailed comparison of ferroelectric properties between **1** and quinuclidinium perrhenate were listed in Table S1 in Supporting Information, also shown below. Description of the novelty of **1** was also included in the introduction part (page 4).

Table S1. Comparison of ferroelectric properties between **1** and Quinuclidinium perrhenate.

	Remnant polarization ($\mu\text{C}/\text{cm}^2$)	Temperature range of ferroelectric phase (K)	Paraelectric phase	Ferroelectric phase	Number of polar-axes
1	6.2 ¹ 6.8 ²	<322	$Pm\bar{3}m$	$Pmn2_1$	6
Quinuclidinium perrhenate ³	3.5	345-367	$Pm\bar{3}m$	$R3m$	4

1 Obtained on thin-film sample on ITO/glass substrate

2 Obtained on thin-film sample on flexible ITO/PET substrate

3 Harada, J. *et al.* Directionally tunable and mechanically deformable ferroelectric crystals from rotating polar globular ionic molecules. *Nature Chemistry*, doi:10.1038/nchem.2567 (2016).

In ref.17, Harada et al. raised a problem for normal molecular ferroelectric

crystals: “restricted orientational control over molecular crystals has severely limited their applications in electronic devices.” Therefore, they synthesized and characterized quinuclidinium perrhenate, in which the polarization axis can be easily changed to the desired direction through rotation of the constituent ionic molecules of the crystal. However, they have not investigated thin films. Therefore, the present study should focus more on the characterization of ferroelectric thin films.

Author’s response:

We do agree that the thin-film study is the significance of our work. Since **1** is a new compound which is first timely reported, some general characterizations on the basic properties are also necessary. Besides those tests, all ferroelectric characterization were performed on thin-film samples, including macroscopic P-E dependence, microscopic domain imaging, microscopic polarization reversal, microscopic poling, microscopic optical imaging, AFM imaging, etc. The macroscopic P-E hysteresis studies were even performed on both ITO/glass and flexible ITO/PET substrates to demonstrate the application potential of **1** in the form of poly-crystalline thin-films.

The six-fold rotational polar-axis is another major concern. Single crystal analysis shows that the quinuclidinium periodate has a six-fold rotational polar-axis. This suggests that an identical P-E curve should be observed when an electric field is applied from any of the six directions; however, it is not directly demonstrated by experiment. The paper could be strengthened by performing P-E curve measurements using single crystal in at least two different directions.

Author’s response:

The reviewer made an excellent suggestion on demonstration of the six-fold rotational polar-axes by measuring macroscopic *P-E* loop in different directions. However, for measuring single crystal sample, the relative large coercive field (255 kV/cm) limits the dimension of sample to be much smaller than 100 μm ,

with highest voltage of 2 kV we can supply. Thus it would be extremely difficult to perform the measurement along two different directions on a <100 μm crystal. On the other hand, since poly-crystalline thin-film samples were prepared without any control of crystallinity, the crystallographic orientations are random over the thin-film. By successfully obtained nice P-E hysteresis loop and observation of polarization reversal on thin-film samples, we have already demonstrated that the macroscopic ferroelectricity can be observed along different crystallographic direction.

Minor issues

Color and atoms should be correlated in Fig. 1.

Figures 2a and 2b should be plotted for the same temperature range, for example, between 280 and 360 K to enable comparison.

Color in Fig. 6 (Phase) appears to be red, and not purple.

In the Berry phase calculation section, “ λ in Figure 3” should be revised to “ λ in Figure S1.”

It would be helpful to provide the definition of λ in Fig. S1.

Label for the y-axis should be added to Fig. S2.

“Heigt” in Fig. S7b should be revised to “Height.”

Author’s response:

Fig. 2a and 2b have been replotted with same temperature range. A brief explanation of dimensionless parameter λ has also been added in the Supporting Information in the section of “Berry phase calculation”, on Page 2. All minor

mistakes have been corrected and the manuscript was carefully proofed again to avoid further mistakes and misleadings.

Reviewer #2 (Remarks to the Author):

This work discovered the new molecular ferroelectrics of quinuclidinium periodate, which can be readily prepared on various substrates without the concern on the film crystallinity and crystal-orientation. The multi-axes (six polar axes) molecular ferroelectrics exhibits the dendritic growth pattern and 12-fold polarization direction. As a result, the thin films with switchable polarization possess great application potentials. There are a great many of outstanding points of this work, such as the simple preparation of aqueous solution processing and the flexible structure on ITO-PET electrode substrate. In addition, the micrometer scale thin-film can be fabricated on various substrates, including the flexible polymers with high uniformity. I strongly recommend this work for the publication.

Author's response:

The authors would like to express our appreciation for reviewer's positive comments on the scientific significance of our work. The reviewer's suggestions are very valuable and helpful for us in the aspect of improving the quality of our manuscript. Below, we response to the comments in blue color.

There are two minor comments for the authors' consideration: 1) It may be helpful to add the rational of smaller remnant polarization (currently discussed in the supporting information as compared to the theoretically predicted one) into the main text. 2) The author may consider add a few sentences to discuss the strategies to reduce the coercive field of the films, though six polar axes of quinuclidinium periodate.

Author's response:

1) The discussion of over-estimated polarization value has been moved from Supporting Information to the main text, following reviewer's suggestion.

- 2) A description of controlling film thickness to reduce coercive potential has been added to the “Macroscopic ferroelectricity on polycrystalline thin-films” section on Page 9, following reviewer’s suggestion.

Point-by-point response

(Our response are marked in blue)

REVIEWERS' COMMENTS:

Reviewer #1 (Remarks to the Author):

The manuscript has been revised appropriately on the basis of two reviewers' comments. The manuscript is worthy for publication after addressing the following issues.

Minor issues

Color and atoms should be correlated in Fig. 1; N (blue) is not included in the revised Figure (Fig. 1).

It would be useful to include the elemental analysis (CHN) of 1 in the Materials section.

Reply: We thank the reviewer for the positive assessment and the comments. The figure legend was updated, and the N atom is included in Fig. 1a. We performed the elemental analysis, and the data are included in the method.

Reviewer #2 (Remarks to the Author):

The authors have addressed and revised the manuscript, which is ok for the publication.

Reply: We thank the reviewer for the positive assessment and the comments.